# Transcriptomic and Proteomic Analysis of Marine Nematode *Litoditis marina* Acclimated to Different Salinities

**DOI:** 10.3390/genes13040651

**Published:** 2022-04-07

**Authors:** Yusu Xie, Liusuo Zhang

**Affiliations:** 1CAS and Shandong Province Key Laboratory of Experimental Marine Biology, Institute of Oceanology, Chinese Academy of Sciences, Qingdao 266071, China; yusuxie@qdio.ac.cn; 2Laboratory of Marine Biology and Biotechnology, Qingdao National Laboratory for Marine Science and Technology, Qingdao 266237, China; 3Center for Ocean Mega-Science, Chinese Academy of Sciences, 7 Nanhai Road, Qingdao 266071, China

**Keywords:** salinity, euryhalinity, hyposaline, hypersaline, marine nematode, *Litoditis marina*

## Abstract

Salinity is a critical abiotic factor for all living organisms. The ability to adapt to different salinity environments determines an organism’s survival and ecological niches. *Litoditis marina* is a euryhaline marine nematode widely distributed in coastal ecosystems all over the world, although numerous genes involved in its salinity response have been reported, the adaptive mechanisms underlying its euryhalinity remain unexplored. Here, we utilized worms which have been acclimated to either low-salinity or high-salinity conditions and evaluated their basal gene expression at both transcriptomic and proteomic levels. We found that several conserved regulators, including osmolytes biosynthesis genes, transthyretin-like family genes, V-type H^+^-transporting ATPase and potassium channel genes, were involved in both short-term salinity stress response and long-term acclimation processes. In addition, we identified genes related to cell volume regulation, such as actin regulatory genes, Rho family small GTPases and diverse ion transporters, which might contribute to hyposaline acclimation, while the glycerol biosynthesis genes *gpdh-1* and *gpdh-2* accompanied hypersaline acclimation in *L. marina*. This study paves the way for further in-depth exploration of the adaptive mechanisms underlying euryhalinity and may also contribute to the study of healthy ecosystems in the context of global climate change.

## 1. Introduction

Salinity is an important abiotic environmental factor, which affects survival, growth, reproduction and ecological distribution of living organisms. Efficient sensation, response and adaption to their external salinity environment is vital for all living individuals. The imbalance of salt intake also affects human health, which is associated with a variety of cardiovascular diseases and other physiological pathologies [1,2,3]. Therefore, osmoregulation mechanisms have always been an important part of biology.

In the process of response and adaptation to salinity changes in the surrounding environment, certain universal strategies are applied by different organisms, yet with diversity in the details of regulation. Studies using brewer’s yeast as a model, have demonstrated that osmotic sensation and transduction within a single eukaryotic cell can be highly complex, acting in parallel pathways, and often cross-communicate with other signaling processes [4,5]. Yeast cells accumulate glycerol as a compatible osmolyte under hyperosmotic stress, and the high osmolarity glycerol (HOG) response pathway controls glycerol accumulation at various levels including the activation of mitogen-activated protein kinase (MAPK) pathway genes [5,6,7]. Upon acute osmotic shocks, cell volume changes rapidly. Hyperosmotic shock causes cell shrinkage, while hypoosmotic stress leads to cell swelling. Along with volume perturbation, the biophysical properties of the cell membrane, physiological state in the cytosol, and DNA structure as well as gene expression in the nucleus will be affected [6,8]. It is known that the actin cytoskeleton [9,10,11,12]; signaling pathways such as Rho family small GTP binding proteins [10,13,14] and MAPK [15,16,17]; membrane transporters such as water channel aquaporins [18,19,20,21] and a variety of sodium, chloride- and potassium-related ion channels [8,22,23,24] together with organic osmolytes such as glycerol, myo-inositol, taurine and methylamines [25,26,27] play critical roles in the process of osmotic regulation. These components are also involved in ensuing adaptive regulation of cells under long-term osmotic stresses [6,8]. In multicellular organisms with more complicated structures, osmoregulation is conducted mainly by osmoregulatory tissues and organs, for instance, gills in crustaceans and fish, kidneys in mammals, which are even regulated by their neuroendocrine systems [28,29,30,31,32,33]. Moreover, disruption to important members of osmoregulatory processes, such as a variety of ion transporters, aquaporins and WNK kinases, has been reported to contribute to diverse human diseases [28,34,35,36,37]. However, the precise mechanisms of osmotic sensation, signal transduction and adaptation remain poorly defined in invertebrate animals.

Marine nematode *Litoditis marina* has emerged as an excellent invertebrate system for osmoregulation studies; it is a euryhaline nematode, inhabiting widely in the littoral zone of coasts and estuaries all over the world [38,39,40]. In nematodes, the hypodermis and its outer cuticle; the excretory system, which is composed of the H-shaped excretory cell, the duct cell and the pore cell; as well as the intestine, are important osmoregulatory tissues, as is well documented in studies on *Caenorhabditis elegans* [41,42]. Extensive studies in *C. elegans* as well as some extremophilic nematodes have revealed a set of important osmoregulatory genes in nematodes, for instance, metabolic genes related to synthesis and accumulation of osmolytes glycerol and trehalose [43,44,45,46]; ion transport-related genes, such as transient receptor potential cation channel (TRP)-family genes and chloride channel genes [42]; aquaporin water channel genes [47,48]; extracellular proteins including some collagens and *osm* genes [42,43,46,49,50,51,52] as well as genes related to MAPK, WNK-1/GCK-3, Notch and insulin-like signaling pathways [42,45,46,52,53,54,55]. However, osmoregulatory studies in the terrestrial nematode *C. elegans* were carried out mostly under hyperosmotic conditions. With a simple body structure, a short lifecycle, an annotated reference genome and applicable gene editing manipulation in *L. marina*, this euryhaline marine nematode can thus be used as an experimental system in studying regulatory and adaptive mechanisms underlying both hyperosmotic and hypoosmotic conditions to delineate the molecular mechanisms underlying euryhalinity.

We have previously identified a broad range of salinity-responding genes in *L. marina* when challenged with either low or high salinity stresses [39]. We demonstrated that transthyretin-like family genes and heat-shock protein genes were presumably general regulators involved in *L. marina*’s damage control mechanism in response to different salinity stresses. Particularly, unsaturated fatty acid biosynthesis-related genes and certain cytoskeleton-related genes probably play an important role in response to hyposaline stress, whereas glycerol biosynthesis genes and cuticle-related collagen genes were involved in the hypersaline stress response [39].

To further explore the adaptive mechanisms underlying its euryhalinity, *L. marina* worms were acclimated to either lower or higher salinity culture conditions, and transcriptomic and proteomic analyses were then performed to study the basal mRNA and protein differences among worms acclimated to different salinity conditions. Nowadays, gradual climate change has already accelerated rises in sea level, which will lead to decreasing ocean salinity while increased salinization of coastal areas, as a result, more species will encounter either hyposaline or hypersaline stresses. This research will not only pave the way for further in-depth exploration on adaptive mechanisms underlying euryhalinity, but also will provide insights into protection and administration of ecosystems which are stressed by gradual climate change.

## 2. Materials and Methods

### 2.1. Worms

Worms acclimated to seawater salinity environment (S30 group): The wild strain of marine nematode *L. marina*, HQ1, was cultured on seawater-NGM (SW-NGM) agar plates prepared with filtered natural seawater (Appendix A), as described previously [38,39]. The salinity of seawater used for this condition was 3%. The *Escherichia coli* strain OP50 was applied as a food source. Worms cultured under this condition were propagated for about 3.5 years before this study.

Worms acclimated to low salinity environment (S3 group): HQ1 worms were transferred from SW-NGM agar plates to normal NGM agar plates containing 0.3% NaCl (Appendix A), which is the standard culture condition for the terrestrial nematode *C. elegans*. The *E. coli* OP50 was applied as a bacterial food source. Worms cultured under this condition were propagated for about 1.5 years before this study.

Worms acclimated to a higher salinity environment (S50 group): HQ1 worms were transferred from SW-NGM agar plates to the artificial seawater-NGM (ASW-NGM) agar plates containing 5% sea salt (Appendix A). ASW-NGM agar plates were prepared by artificial sea salt (Instant Ocean). *E. coli* OP50 was applied as the food source. Worms cultured under this condition were propagated for about 3 months (more than 10 generations) before this study.

All worms were maintained at 20 °C in the laboratory.

### 2.2. Developmental Analysis under Different Salinity Conditions

For developmental analysis, 100 newly hatched *L. marina* L1 larvae were transferred onto each indicated 35 mm diameter agar plate, seeded with 30 μL OP50. The number of adult worms was scored every 24 h.

Data represent the average of three replicates. Statistical significance was determined using the two-tailed Student’s *t*-test between two groups. *p* value < 0.05 was considered statistically significant.

For egg laying time analysis, the earliest observed egg laying time on each assay plate was recorded, 100 L1 larvae per plate in six replicates for each condition.

### 2.3. Lifespan Assay

Worms were cultured under optimal growth conditions for at least 3 generations before lifespan assay. The lifespan assay was performed starting at day 1 of adulthood as described previously [38,56], with minor modification. In brief, 35 mm diameter assay plates seeded with 30 μL OP50 were prepared every day. Forty L4 females were transferred to each assay plate, and incubated at 20 °C. The number of live and dead worms was determined using a dissecting microscope every 24 h. Alive worms were transferred to fresh OP50-seeded agar plates daily. Three replicates were analyzed. Worms were scored as dead if no response was detected after prodding with a platinum wire. Dead worms on the wall of the plate were not counted.

Statistical analysis of the average lifespan for worms acclimated to each salinity condition was performed. Data represent the average of three replicates. The comparisons between two groups were performed using the two-tailed Student’s *t*-test. *p* value < 0.05 was considered statistically significant.

### 2.4. Synchronized L1 Larvae Collection for Each Salinity Condition

The synchronized L1 larvae under each salinity condition were collected as previously reported by Xie et al. [39]. Worms, acclimated to S3, S30 and S50 conditions, were allowed to lay eggs overnight at 20 °C. Eggs were washed off and collected using corresponding suitable solutions, specifically, M9 buffer for the S3 group, filtered sterile seawater for the S30 group, and filtered sterile artificial seawater containing 5% sea salt (Instant Ocean) for the S50 group. The eggs were then treated with worm bleaching solution (Sodium hypochlorite solution:10 M NaOH:H_2_O = 4:1:10, prepared in terms of volume ratio) at room temperature for 1.5 min. Eggs were then washed twice with the corresponding suitable solution. The clean eggs hatched overnight and developed into L1 larvae in the corresponding suitable solution at 20 °C. The synchronized L1 larvae were collected by filtration using a 500-grid nylon filter with 25 μm mesh size. The samples were frozen immediately in liquid nitrogen.

### 2.5. RNA-Seq Analysis

Total RNA was extracted using Trizol (Invitrogen, Carlsbad, CA, USA). With three biological replicates for each group, a total of nine RNA libraries were prepared with 3 μg RNA using NEBNext^®^ UltraTM RNA Library Prep Kit for Illumina^®^ (New England Biolabs, Ipswich, MA, USA) following the manufacturer’s instructions. Then, RNA libraries were sequenced on an Illumina NovaSeq 6000 platform and 150 bp paired-end reads were generated.

First, clean data were obtained by removing reads containing sequencing adaptors, reads having poly-N and low-quality ones from raw data. The minimum of base score Q20 was over 97.79% and Q30 was over 93.64%. Then, the clean data were aligned to the *L. marina* reference genome [38] by Hisat2 (v2.0.5, with the default parameters) [57], with mapping ratio from 68.57% to 70.70% (Appendix A). New transcripts for novel genes were predicted and assembled by StringTie (v1.3.3b, with the default parameters) [58], then annotated with Pfam, SUPERFAMILY, Gene Ontology (GO) and KEGG databases. Briefly, the functional annotation was performed using InterProScan v5.17-56.0 [59] by searching against publicly available databases Pfam (http://pfam.xfam.org/, accessed on 27 July 2020), SUPERFAMILY (http://supfam.org, accessed on 27 July 2020), and GO (http://www.geneontology.org/, accessed on 27 July 2020), with an E value cutoff of 1 × 10^−5^. KEGG function [60] was assigned using KOBAS 3.0 [61] by best hit (with an E value cutoff of 1 × 10^−5^) to KEGG database (http://www.genome.jp/kegg/, accessed on 27 July 2020). Further, the read numbers mapped to each gene were analyzed using featureCounts (v1.5.0-p3, with parameter -Q 10 -B -C) [62], and FPKM (expected number of fragments per kilobase of transcript sequence per million base pairs sequenced of each gene) was calculated based on the length of the gene and read counts mapped to this gene, which was used for estimating gene expression levels. Hierarchical clustering for 9 samples performed using the pheatmap package in R and shown in Appendix A indicated the sample preparation was reliable. Subsequently, differential expression analysis of two conditions was performed using the DESeq2 R package (v1.16.1) [63]. The resulting *p* values were adjusted using the Benjamini and Hochberg approach for controlling the false discovery rate. Genes with an adjusted *p* value (padj) < 0.05 found by DESeq2 were assigned as differentially expressed. Moreover, GO enrichment analysis and KEGG pathway enrichment analysis for differentially expressed genes (DEGs) were achieved by clusterProfiler R package (v3.4.4), an adjusted *p* value (padj) < 0.05 was considered significantly enriched. GeneRatio was defined as the ratio of the number of differential genes annotated to the GO term or on the KEGG pathway to the total number of differential genes, respectively.

### 2.6. Proteomic Analysis

Worm samples were sonicated three times on ice using a high intensity ultrasonic processor (pulsed at 25% power for 3 s on and 5 s off for 3 min, Scientz) in lysis buffer (8 M urea, 1% protease inhibitor cocktail), with three biological replicates for each group. The remaining debris was removed by centrifugation at 12,000× *g* at 4 °C for 10 min. Then, the supernatant was collected, and the protein concentration was determined with a BCA kit (Beyotime Biotechnology, Shanghai, China) according to the manufacturer’s instructions. Protein samples were digested with trypsin (Promega, Madison, WI, USA) at 1:50 trypsin-to-protein mass ratio overnight. After being desalted by Strata X C18 SPE column (Phenomenex, Torrance, CA, USA) and vacuum-dried, peptides were labelled with a tandem mass tags (TMT) kit (ThermoFisher Scientific, Waltham, MA, USA) according to the manufacturer’s protocol.

Next, the TMT-labeled peptides were fractionated by high-pH reverse-phase HPLC using Agilent 300 Extend-C18 column (5 μm particles, 4.6 mm ID, 250 mm length; Agilent, Santa Clara, CA, USA). Briefly, peptides were first separated with a gradient of 8% to 32% acetonitrile (pH 9.0) over 60 min into 60 fractions. Then, the peptides were combined into 9 fractions and dried by vacuum centrifugation.

Further, peptides were identified and quantified by liquid chromatography-tandem mass spectrometry (LC-MS/MS). Briefly, the tryptic peptides were dissolved in solvent A (0.1% formic acid in 2% acetonitrile), directly loaded onto a home-made reversed-phase analytical column (75 μm ID, 15 cm length). The gradient comprised an increase from 9% to 22% solvent B (0.1% formic acid in 90% acetonitrile) over 40 min, 22% to 32% in 14 min and climbing to 80% in 3 min then holding at 80% for the last 3 min, all at a constant flow rate of 450 nL/min on an EASY-nLC 1200 UPLC system (ThermoFisher Scientific). The peptides were subjected to a nanospray ionization (NSI) source followed by tandem mass spectrometry (MS/MS) in Q Exactive HF-X (ThermoFisher Scientific) coupled online to the UPLC. The electrospray voltage was set as 2.2 kV. The *m*/*z* scan range was 350 to 1400 for full scan, and intact peptides were detected in the Orbitrap at a resolution of 120,000. Peptides were then selected for MS/MS using the normalized collisional energy (NCE) setting at 28 and the ion fragments were detected in the Orbitrap at a resolution of 30,000. A data-dependent procedure that alternated between one MS scan followed by 20 MS/MS scans with a 15 s dynamic exclusion was used. Automatic gain control (AGC) was set at 1 × 10^5^. The fixed first mass was set as 100 *m*/*z*.

The resulting MS/MS data were processed using the Maxquant search engine (v1.5.2.8). Tandem mass spectra were searched against the 17,661 protein database of *L. marina* [38] concatenated with the reverse decoy database. Trypsin/*p* was specified as the cleavage enzyme allowing up to 2 missing cleavages and 5 modifications per peptide. The mass tolerance for precursor ions was set as 10 ppm in the first search and 5 ppm in the main search, and the mass tolerance for fragment ions was set as 0.02 Da. Carbamidomethyl on Cys was specified as fixed modification, and acetylation on protein N-terminal, oxidation on Met, deamidation on Asn and Gln were specified as variable modifications. Minimum peptide length was set at 7. The quantitative method was set to TMT 10plex, and the false discovery rate (FDR) for protein identification was adjusted to < 1%. All the other parameters in MaxQuant were set to default values. A *p* value < 0.05 from *t*-tests and a fold change > 1.3 or < 1/1.3 were set as the thresholds for significantly differentially expressed proteins (DEPs).

A total of 306,324 spectrums were acquired, of which 78,015 unique spectrums were obtained, and a total of 45,669 peptides were identified (Appendix A). The length distribution analysis of peptides showed that most of them consisted of 7–20 amino acids, which is in accordance with the quality control requirements (Appendix A). Subsequently, hierarchical clustering for 9 samples are shown in Appendix A, indicated the reliable sample preparation.

Moreover, different databases were selected for protein functional annotation. GO annotation proteome was derived from the UniProt-GOA database (http://www.ebi.ac.uk/GOA/, accessed on 30 July 2020). If the proteins were not annotated by UniProt-GOA database, the InterProScan (https://www.ebi.ac.uk/interpro, accessed on 30 July 2020) would be applied to annotate by protein sequence alignment method, which was also used for protein domain functional description. KEGG online service tools KAAS was utilized to predict the pathways in which the identified proteins were involved. Then the annotation results were mapped on the KEGG pathway database using KEGG mapper. Subcellular localizations of the protein were predicted by wolfpsort (http://wolfpsort.seq.cbrc.jp/, accessed on 30 July 2020). COG annotation of the protein was achieved by eggnog-mapper software (v2.0) with the default parameters.

Additionally, enrichment analysis of GO and KEGG pathway was conducted for DEPs by Python, using an in-house script. Two-tailed Fisher’s exact test was employed to test the enrichment of DEPs against the background of all identified proteins and a corrected *p* value < 0.05 was considered significantly enriched. Ratio was defined as the ratio of the number of differential proteins annotated to the GO term or on the KEGG pathway to the total number of differential proteins, respectively.

## 3. Results

### 3.1. L. marina Is a Typical Euryhaline Marine Nematode

In the laboratory, marine nematode *L. marina* HQ1 is normally cultured on SW-NGM agar plates prepared with seawater with a salinity of 3%, and this condition is referred to as “S30” in this paper. Basic developmental characteristics were tested in the laboratory, including the adulthood percentage, earliest egg laying time and lifespan. Under the S30 condition, although only 3% of newly hatched L1 larvae had developed into the adult stage within 3 days at 20 °C, the adulthood percentage was as high as 92% within a week (Figure 1A). The earliest observed egg laying time was around 84 h post L1 stage (Figure 1B). Moreover, L4 worms lived as long as 29 days (Figure 1C), with an average lifespan of about 16 days (Figure 1D).

Interestingly, HQ1 worms acclimated quite well to the low-salinity condition (0.3% NaCl, referred to as “S3”) which is the standard culture condition for terrestrial *C. elegans*. We observed that, under S3 condition, 2% of L1s developed into adulthood within 3 days, the adulthood percentage was about 92% on the 7th day, and the earliest egg laying time was around 83 h post L1 stage, which showed no significant differences compared with the S30 group (Figure 1A,B). However, worms’ lifespan under S3 were slightly shorter: L4 worms lived as long as 27 days (Figure 1C), with an average lifespan of about 13 days (Figure 1D).

HQ1 worms were able to propagate and acclimate to an even higher salinity environment (5% artificial sea salt, referred to as “S50”). Notably, L1 worms could not develop into the adult stage within 3 days under the S50 condition. The percentage of adulthood on the 4th and the 5th days also showed significant differences between S50 and S30 groups, while no differences were observed afterwards (Figure 1A). The earliest egg laying time for S50 worms was around 98 h post L1 stage, which showed a severe egg laying delay comparing with both S30 and S3 worms (Figure 1B). L4 worms acclimated to the S50 condition could live as long as 29 days (Figure 1C), with an average lifespan of about 15 days (Figure 1D), which was similar to that of S30 worms.

Taken together, HQ1 worms could acclimate to a wide range of salinity and are one typical euryhaline marine nematodes. How the worms regulate gene expression to acclimate to different salinity environments, remains unknown. Next, we applied newly hatched L1s to investigate the basal differences at both transcriptomic and proteomic levels, respectively.

### 3.2. Analysis of the Basal Transcriptome for L. marina Acclimated to Different Salinity Environments

To investigate the basal transcriptomic differences of *L. marina* growing under different salinity environments, we applied newly hatched L1s for RNA-seq analysis. Significant DEGs were identified from different comparison groups (Figure 2A), with |log_2_foldchange| > 1 and DESeq2 padj < 0.05 setting as the differential gene screening thresholds. Details of these DEGs were listed in Appendix A.

In the low-salinity S3 group, a total of 1191 genes were significantly up-regulated while 773 genes were down-regulated when compared with S30 group (Figure 2A). Based on GO enrichment analysis, significant enrichment was only observed within down-regulated DEGs. Such genes were mainly annotated to “chromosome”, “cellular macromolecular complex assembly”, “cellular component biogenesis”, “intracellular non-membrane-bounded organelle”, “plasma membrane” and “extracellular matrix” (Figure 2B). Moreover, “beta-alanine metabolism” pathway-related genes were significantly enriched among up-regulated DEGs via KEGG pathway enrichment analysis (Appendix A).

There were relatively fewer DEGs identified in the S50 versus S30 comparison group compared with the S3 versus S30 comparison group (Figure 2A). Compared with the S30 group, 114 genes were significantly up-regulated in S50 group, of which “purine metabolism” pathway related genes were significantly enriched (Appendix A). In addition, there were 185 DEGs exhibiting significant down-regulation. We found that genes annotated to “chromosome”, “cellular component biogenesis”, “intracellular organelle part”, “intracellular non-membrane-bounded organelle”, “serine-type endopeptidase inhibitor activity”, “structural constituent of cuticle” and other GO terms were significantly enriched within these down-regulated DEGs (Figure 2B).

Additionally, a comparison between the S3 and S50 groups was also conducted. There were 1346 DEGs up-regulated in the S3 group (Figure 2A), of which “signaling”-related genes were significantly enriched via GO enrichment analysis (Figure 2B). It was interestingly to note that a set of small GTPase related genes were included (*efa-6*, *exc-5*, *frm-3*, *itsn-1*, *tag-77*, *tiam-1*, *unc-73*, *R05G6.10* and *Y37A1B.17*). On the other hand, 876 genes were significantly up-regulated in S50 group (Figure 2A), of which “glutathione metabolism”, “drug metabolism-cytochrome P450”, “fatty acid degradation”, “fatty acid metabolism” and “metabolism of xenobiotics by cytochrome P450” pathway-related genes were enriched (Appendix A). Further, GO enrichment analysis demonstrated that “oxidoreductase activity” genes, “metallopeptidase activity” genes, “extracellular region” genes (Notably, 9 out of the 13 enriched genes were transthyretin-like family genes, TTLs, such as *ttr-31*, *ttr-32*, *ttr-40*, *ttr-44* and *ttr-51*.), “heme binding” genes, “cofactor binding” genes, “peptidase activity” genes, “iron ion binding” genes, “peroxisome” genes and others were significantly enriched within up-regulated DEGs in the S50 group (Figure 2B).

### 3.3. Analysis of the Basal Proteome for L. marina Acclimated to Different Salinity Environments

In parallel, we also applied quantitative proteomic analysis to investigate the basal protein differences among worms growing under different salinity environments. Newly hatched L1s were used, the same developmental staged worms for above transcriptomic analysis. A total of 6068 proteins were identified (Appendix A) and significant DEPs were selected from different comparison groups (Figure 2C), with ratio of fold change > 1.3 or < 1/1.3 and *p* value < 0.05 as the differential protein screening thresholds. Details of these DEPs are listed in Appendix A.

In the low-salinity S3 group, 144 up-regulated proteins and 168 down-regulated proteins were selected, compared with the S30 group (Figure 2C). Among these up-regulated DEPs, ribosome-related proteins were the most significantly enriched ones revealed by both GO enrichment and KEGG pathway enrichment analysis (Figure 2D and Appendix A), others such as “vacuolar membrane” proteins, “dauer larval development” proteins, “isomerase activity” proteins, “negative regulation of mitotic cell cycle” proteins, “nematode larval development” proteins, “positive regulation of catabolic process” proteins and “phosphatidylinositol phosphate binding” proteins were also significantly enriched (Figure 2D). However, “extracellular region” proteins were the most significantly enriched ones in down-regulated proteins. Besides, cytoskeleton related proteins (Appendix A), defense response related proteins, “DNA packaging” proteins, “modified amino acid binding” proteins, “metal ion binding” proteins and “divalent inorganic cation transport” proteins were also significantly enriched (Figure 2D).

Compared with the S30 group, there were 56 proteins significantly up-regulated in the S50 group (Figure 2C), of which ribosome-related proteins were also significantly enriched via both GO enrichment and KEGG pathway enrichment analysis (Figure 2D and Appendix A). Moreover, “chromosomal region” proteins, “midbody” proteins (Appendix A) and “phosphoric ester hydrolase activity” proteins were also significantly enriched among these up-regulated DEPs (Figure 2D). Additionally, 105 down-regulated DEPs were found in the S50 group (Figure 2C). Proteins related with “extracellular region” exhibited the most significant enrichment similar to the low-salinity S3 group. Microtubule-related proteins (Appendix A), defense response related proteins and “hydrolase activity, acting on ester bonds” proteins were also significantly enriched among down-regulated DEPs (Figure 2D).

In addition, we also performed a comparison between the S3 and S50 groups, and found 154 up-regulated DEPs in the S3 group and 143 in the S50 group (Figure 2C). Based on KEGG pathway enrichment analysis, “lysosome” pathway-related proteins and “DNA replication” pathway-related proteins were the most significantly enriched ones in the S3 and S50 groups, respectively (Appendix A). Besides, based on GO enrichment analysis, “response to nicotine” proteins, “proton-transporting V-type ATPase complex” proteins (including VHA-5 and VHA-6), “whole membrane” proteins, “regulation of cellular catabolic process” proteins, “spindle microtubule” proteins (Appendix A), “negative regulation of mitotic cell cycle” proteins, “regulation of proteolysis” proteins, “regulation of synapse organization” proteins exhibited significant enrichment in the S3 group (Figure 2D). In contrast, “MCM complex” proteins, “condensed chromosome” proteins, “extracellular region” proteins, “cation transport” proteins, “nucleotide metabolic process” proteins, “cell division site” proteins, “hydrolase activity, acting on glycosyl bonds” proteins and others were significantly enriched in the S50 group (Figure 2D).

### 3.4. Identification of Genes Expressed Consistently at Both mRNA and Protein Levels in Different Salinity Environments

Interestingly, not only was the salinity difference between the S50 and S30 conditions smaller than that between the S3 and S30 conditions, but we also noticed a similar tendency in the number of DEGs and DEPs identified within different comparisons (Figure 2A,C), i.e., the difference between the S50 and S30 groups was smaller than that between the S3 versus S30 comparison pair, especially for DEGs. Herein, we will refer to these worms which were acclimated to S3 environment as the “low-salinity group”, while worms grown under S30 and S50 environments as the “high-salinity groups”. In order to identify crucial genes related to euryhalinity, we tried to screen genes that were expressed consistently at both mRNA and protein levels.

As shown in Figure 3A, 1638 genes were up-regulated specifically in the low-salinity group (S3), compared with the high-salinity groups (S30 and S50), with the screening thresholds set to fold change > 1.0, padj < 0.05 applied to the transcriptomic data. Similarly, with the screening thresholds set to fold change > 1.0, *p* value < 0.05 applied to the proteomic data, a total of 354 proteins were specifically up-regulated in the low-salinity group. Furthermore, we combined the results from transcriptomic and proteomic analysis and found that 78 genes exhibited consistent expression at both mRNA and protein levels. Interestingly, we also found that 29 out of the above 78 genes, including the trehalose biosynthesis gene *tps-2*, the transthyretin-like family genes *ttr-15* and *ttr-30*, the ion transport genes *mca-1*, *twk-33* and *vha-5*, demonstrated specific induction upon hyposaline stress in *L. marina* based on our previous data (Appendix A). Thus, these 78 genes were considered low-salinity-specific genes, and their detailed information is summarized in Table 1.

To identify high-salinity-specific genes, similar analysis was performed as described above. Briefly, 1015 genes and 336 proteins were specifically up-regulated in high-salinity groups (S30 and S50), respectively (Figure 3B). A total of 69 genes were further selected as high-salinity-specific genes, with detailed information summarized in Table 2. Additionally, there were 11 genes among these 69 genes, including the glycerol biosynthesis gene *gpdh-1* and the cuticle related collagen gene *col-160*, can be specifically induced by hypersaline stress (Appendix A).

Further, the differential genes expressed specifically under low- and high-salinity conditions were classified based on their annotation information (Figure 3C). We found that, under the low-salinity condition, the top six gene categories with annotated functions were grouped into “signal transduction mechanisms” (*arr-1*, *mnk-1*, *ppfr-1* and *tnc-2*), “nucleotide transport and metabolism” (*alh-3*, *ent-1*, *hint-3*, *pnp-1* and *pyr-1*), “cytoskeleton” (*ben-1*, *gsnl-1*, *tni-3* and *tnt-3*), “lipid transport and metabolism” (*cest-26*, *faah-2* and *ltah-1.1*) and “transcription” (*dmd-7*, *gmeb-2*, *isw-1*, *ldb-1* and *sta-1*) (Figure 3C and Table 1). However, “posttranslational modification, protein turnover, chaperones” (*cpr-5*, *dpy-31*, *fkb-2*, *gst-8* and *rle-1*), “amino acid transport and metabolism” (*alh-6* and *alh-9*), “lipid transport and metabolism” (*acox-1.6*, *acs-7* and *ges-1*), “defense mechanisms” (*clec-48*, *clec-49*, *cri-3* and *mpst-1*), “energy production and conversion” (*alh-11*, *gpdh-1* and *gpdh-2*) and “signal transduction mechanisms” (*lrp-1*, etc.)-related genes were the top six categories under high-salinity conditions (Figure 3C and Table 2).

### 3.5. Identification of Genes and Their Corresponding Proteins, the Abundance of Which Was Proportional to Environmental Salinity

In the present study, we found that DEGs enriched markedly based on RNA-seq analysis, hardly exhibited enrichment at the protein level (Figure 2B,D), in fact, only limited genes were expressed consistently at both mRNA and protein levels (Figure 3A,B). Next, we focused on those genes whose abundance was directly or inversely proportional to environmental salinity at both mRNA and protein levels, which were probably key regulators or effectors associated with the effective osmoregulation of euryhaline *L. marina*.

We therefore combined transcriptomic and proteomic profiles and further found that 66 genes and their corresponding proteins demonstrated environmental salinity-dependent patterns in expression (Appendix A). Specifically, 38 genes were down-regulated when salinity increased, while 28 genes were up-regulated when salinity increased (Figure 4 and Appendix A).

We classified the candidates by their function annotation information and found that, interestingly, “cytoskeleton”-related genes (*ben-1*, *gsnl-1*, *tni-3*, *tnt-3* and *EVM0008515*), “inorganic ion transport and metabolism”-related genes (*kcc-1*, *mca-1* and *twk-33*), “signal transduction mechanisms”-related genes (*mnk-1*, *ppfr-1* and *tnc-2*), “transcription”-related genes (*dmd-7*, *gmeb-2* and *isw-1*) and others (“intracellular trafficking, secretion, and vesicular transport”, “nucleotide transport and metabolism” and “secondary metabolites biosynthesis, transport and catabolism”) were notably grouped, showing up-regulation with decreasing salinity (Figure 4A,B). On the contrary, “energy production and conversion”-related genes (*alh-11*, *gpdh-1*, *gpdh-2* and *Y71G12B.10*), “lipid transport and metabolism”-related genes (*acox-1.6*, *acs-7* and *ges-1*), “posttranslational modification, protein turnover, chaperones”-related genes (*gst-8*, *rle-1* and *Y71H2AM.1*), and others (“extracellular structures” and “defense mechanisms”) were grouped, exhibiting up-regulation with increasing salinity (Figure 4A,B).

Together, the above genes could be key regulators or effectors in *L. marina*, involved in its acclimation process to different salinity environments, and worthy of further in-depth functional study in the future.

## 4. Discussion

Effective osmotic regulation not only directly affects animals’ survival, but also shapes their behaviors and distribution. There are various euryhaline fish in nature, such as eels, which spending most of their lives in freshwater until they return to their spawning grounds in the sea, whereas salmon migrate from ocean through their natal river for spawning [64,65,66]. During freshwater to seawater transition or vice versa, euryhaline fish although cope with external salinity changes in a species-specific way, evolutionary conserved strategies do exist among them. The gill, intestine and kidney are the major osmoregulatory tissues in fish and successful acclimation in both freshwater and seawater environments depends on proper physiological, metabolic and structural adjustments within these tissues, which also involves neuroendocrine regulation [29,67]. Euryhaline fish have the ability to perceive salinity changes through multiple osmo-sensors, including transmembrane proteins and cytoskeletal proteins, which results in early osmotic response and regulation processes and the following regulatory expression and activities of diverse genes and corresponding proteins involved in water- and ion transport, macromolecular damage control, accumulation and transport of organic osmolytes and other physiological processes together contribute to cellular stress response, cell volume regulation and tissue remodeling [64,67,68,69,70]. In addition to the final physiological acclimation to external salinity, there are also phenotypic differences in behavior and body morphology or size between marine and freshwater populations, as reported in *Fundulus* and three-spined sticklebacks [71,72]. Similar regulatory mechanisms have also been reported in crustaceans, such as crabs and shrimps [30,31,73].

When animals encounter a stress-inducing habitat, they will first trigger early stress responses and then might adapt to the new environment through subsequent adaptive regulation. Our previous report showed that *L. marina* L1 larvae were paralyzed immediately upon both low- and high-salinity conditions, and the stressed worms exhibited developmental defects afterwards [39]. In the present study, we observed that *L. marina* could move, grow and propagate normally after acclimation to either hyposaline or hypersaline environments, except for a relatively shortened lifespan for hyposaline-acclimated worms and relatively delayed development of hypersaline-acclimated worms. We then analyzed the basal transcriptomic and proteomic differences among *L. marina* worms acclimated to different salinities, aiming to provide insight into invertebrate euryhalinity.

### 4.1. Cellular Stress Response Genes Might Play Essential Roles in L. marina Euryhalinity

Several basic aspects of the osmotic-induced cellular stress response are well documented, including regulation of the cell cycle and reallocation of metabolic energy [70]. Here, diverse genes related to “cell cycle control, cell division, chromosome partitioning” and “replication, recombination and repair”, were identified in both low- and high-salinity-acclimated groups, as summarized in Figure 3C. Similarly, metabolism-related genes associated with “amino acid transport and metabolism”, “carbohydrate transport and metabolism”, “coenzyme transport and metabolism”, “lipid transport and metabolism”, “nucleotide transport and metabolism” and “energy production and conversion” also exhibited differential expression patterns in worms growing under hyposaline and/or hypersaline-acclimated conditions (Figure 3C). Interestingly, we found that many of these genes, especially the metabolism-related ones, could also be induced by early short-term salinity stresses [39], for instance, *bus-2*, *gdh-1*, *pnp-1*, *tatn-1*, *C01B10.3*, *F08F3.4*, *Y43F8C.13* and *Y71F9B.9* were responsive to hyposaline stress (Table 1 and Appendix A), while *gpdh-1*, *F37C4.6* and *Y71G12B.10* were responsive to hypersaline stress (Table 2 and Appendix A), suggesting the involvement of these genes in both osmotic stress response and acclimation processes in euryhaline *L. marina*.

Another aspect of the osmotic-induced cellular stress response is programmed cell death [70]. Previously, we reported a group of TTL genes presumably involved in apoptosis and damage control regulation in response to both low- and high-salinity stresses in *L. marina* [39]. Here, we found that diverse TTL genes were differentially expressed in acclimation to different salinity environments. A total of 31 out of about 47 annotated TTL genes in *L. marina*’s genome exhibited expression differences with significance in at least one comparison group at either mRNA or protein level or both (Appendix A). For instance, 10 TTL genes were significantly increased under the low-salinity condition, 4 of which were increased at both mRNA and protein levels (EVM0016470/*ttr-59*, EVM0008626/*ttr-46*, EVM0003972/*ttr-15* and EVM0004638/*ttr-30*). In contrast, 17 TTL genes were up-regulated under high-salinity condition(s), 4 of which were significantly up-regulated at both mRNA and protein levels (EVM0002004/*ttr-51*, EVM0005297/*ttr-44*, EVM0007550/*ttr-30* and EVM0011170/*ttr-27*). Together, this distinctively differential regulation of diverse TTL genes in either hyposaline- or hypersaline-acclimated nematodes, suggests that certain TTL genes might play essential roles in hyposaline acclimation, while others might play essential roles in hypersaline acclimation. Moreover, 16 out of the above 31 salinity acclimation-related TTL genes were identified in our previous study [39], and could be induced by both hypo- and hyper-osmotic stresses, such as *ttr-30*, *ttr-44*, *ttr-46* and *ttr-59* (Appendix A). As one of the largest nematode protein families, most TTLs were predicted to be secreted [74,75]. Some TTL genes in *C. elegans* were responsive to diverse environmental challenges including oxidative stress, pathogen exposure and osmotic stress [52,76,77,78,79]. According to studies on *ttr-52*, which was reported as a bridging factor involved in cell corpse engulfment, apoptosis and axon repair [75,80,81], and another TTL gene, *ttr-33*, was reported as a potential secreted sensor or scavenger of oxidative stress involved in neuroprotective mechanism [82]. We thus speculated that TTL genes presumably function in apoptosis and damage-control machinery involved in the cellular stress response to cope with salinity stresses and play important roles in both salinity response and acclimation processes in euryhaline *L. marina*. How each TTL gene functions in salinity acclimation deserves to be further explored.

### 4.2. Certain Cell Volume-Regulation Genes Might Contribute to Hyposaline Acclimation in L. marina

Cell volume is known to be affected by osmotic exposure [8,42]. Volume-dependent regulation will be elicited to restore near-normal cell volume to maintain homeostasis after volume perturbation [8]. The cytoskeleton was implicated as a potential volume sensor and a mediator of volume-dependent regulation of various ion transporters and channels [8,83,84]. It has been well documented that cytoskeletal remodeling, especially actin-related, is involved in volume changes upon osmotic stress [11,85]. Previously, we reported that several tubulin genes show significantly opposing transcriptional changes between low- and high-salinity-stressed worms, indicating their potential roles in salinity response in *L. marina* [39]. In the present study, although some tubulin-related proteins were enriched in different salinity comparison groups based on the proteomic data, they were hardly expressed consistently at the mRNA level (Figure 2D and Appendix A). However, five cytoskeleton-related genes, *ben-1*, *gsnl-1*, *tni-3*, *tnt-3* and *EVM0008515*, exhibited remarkable elevation at both mRNA and protein levels in low-salinity-acclimated worms (Table 1). Expression of all these five genes was inversely proportional to the environmental salinity, increasing their abundance as salinity decreases (Figure 4B), suggesting their important roles in hyposaline acclimation in *L. marina*. Three of these are actin-regulatory genes, for example, *tni-3* and *tnt-3* are orthologs of human troponin I and troponin T, respectively [86,87], *gsnl-1* is an ortholog of the human gelsolin-family gene [88,89]. Both troponin and gelsolin are involved in regulation of actin dynamics in a calcium-dependent manner, playing important roles in various actin-related processes, including cell structure, cell growth, cell motility, intracellular transport and muscle contraction [90,91,92]. Notably, to our knowledge, this is the first time a potential role for troponin- and gelsolin-related proteins in hyposaline acclimation has been exhibited in *L. marina*, or even in marine invertebrates.

Interestingly, corresponding to the expression profiles of the above actin-regulatory genes, one plasma membrane calcium pump gene *mca-1* [93,94] also exhibited elevation in both mRNA and protein abundances when salinity decreased (Figure 4B). As an important component for the maintenance of calcium homeostasis in cells, *mca-1* probably has a role in actin cytoskeleton regulation in *L. marina*. Besides, Rho family small GTPases have been suspected to participate in the osmotically induced remodeling of the cytoskeleton [11,84]. Consistent with these findings, we also found a set of small GTPase-related genes via GO enrichment analysis, which only increased their transcriptional levels under the low-salinity condition (Appendix A), indicating their potential involvement in cytoskeleton regulation associated with hyposaline acclimation in *L. marina*.

In marine invertebrates and fish, ion transporters and channels play important roles in osmoregulation [30,31,33,67,95]. One example is the upregulation of V-type H^+^-transporting ATPase genes in response to low-salinity stress, which have been documented not only in crustaceans, such as the shrimp *Litopenaeus vannamei* [96] and the mud crab *Scylla paramamosain* [31], but also in the marine nematode *L. marina* [39], indicating their conserved function among marine invertebrates. Besides, Bonzi et al. [66] reported that ion transport genes showed significant expression differences in gill tissues between two natural populations of Arabian pupfish (*Aphanius dispar*), which inhabit nearly fresh water and seawater areas. In our present study, in addition to the V-type H^+^-transporting ATPase gene *vha-5* and a calcium pump gene *mca-1* discussed above, expression of a potassium channel gene *twk-33* [97] and a potassium/chloride cotransporter gene *kcc-1* [98], was also found inversely proportional to the level of salinity, elevating both mRNA and protein abundances when salinity was decreasing (Figure 4B). The increased abundance of potassium channel and potassium/chloride cotransporter genes was consistent with their general requirement mediating K^+^ and Cl^−^ efflux to defend against cell swelling under hypoosmotic conditions [8,99]. Of note, *mca-1*, *twk-33* and *vha-5* also demonstrated specific induction upon hyposaline stress in our previous report [39] (Table 1 and Appendix A), we therefore proposed that these above-mentioned ion transport-related genes play crucial roles in both osmotic stress response and long-term acclimation processes specifically under low-salinity conditions in *L. marina*.

### 4.3. The Glycerol Biosynthesis Genes gpdh-1 and gpdh-2 Accompanied Hypersaline Acclimation in L. marina

It is known that the accumulation of organic osmolytes is a ubiquitous mechanism in cellular osmoregulation [26,27]. There are a number of organic osmolytes such as glycerol, trehalose, inositol, betaine and taurine, which allow cells to counteract the effects of hyperosmolarity and to adapt to hyperosmotic conditions. For example, free amino acids and methylamines are mainly utilized by most marine invertebrates [30,31], whereas glycerol is the most important osmolyte for the terrestrial nematode *C. elegans* [42,46]. It is well documented that an increased level of glycerol is essential for *C. elegans*’ survival in hypertonic environments mediated by upregulation of the glycerol biosynthetic enzyme gene *gpdh-1* [43,46]. Burton et al. reported that if *C. elegans* was exposed to mild high-salinity stress, its progeny could be protected from strong osmotic stress via increasing the expression of *gpdh-2*, and this glycerol synthesis gene is essentially required for the intergenerational osmotic protection [45]. In line with its terrestrial relative *C. elegans*, we previously demonstrated that *gpdh-1* was significantly induced by high-salinity stress in *L. marina*, suggesting that glycerol as an essential osmolyte in both nematodes’ response to hyperosmotic stress [39]. In the current study, two glycerol synthesis genes, *gpdh-1* and *gpdh-2*, both showed significant up-regulation in high-salinity groups at both mRNA and protein levels (Table 2 and Figure 4B), whose expression was directly proportional to the level of salinity. Given *gpdh-2* was not notably induced by salinity stress, our results indicate that *gpdh-2* might play a role in intergenerational osmotic protection inheritance and hyperosmotic acclimation in the marine nematode *L. marina*. Together, our data suggest that both *gpdh-1* and *gpdh-2* are essential to hypersaline acclimation in *L. marina*.

## 5. Conclusions

In conclusion, we have described for the first time the genome-wide transcriptional and proteomic analysis of the marine nematode *L. marina* acclimated to either low- or high-salinity conditions. We found that various cellular stress response genes may function as the conserved regulators in both short-term salinity stress response and long-term acclimation processes. Additionally, we identified diverse genes involved in cell volume regulation which might contribute to hyposaline acclimation, and we also demonstrated that genes related to glycerol biosynthesis might accompany hypersaline acclimation in *L. marina*. Thus, our data might lay the foundation to identify the key gene(s) for further in-depth exploration on environmental adaptation mechanisms in euryhaline organisms, especially in the context of global climate change and the corresponding marine salinity stresses.

## Figures and Tables

**Figure 1 genes-13-00651-f001:**
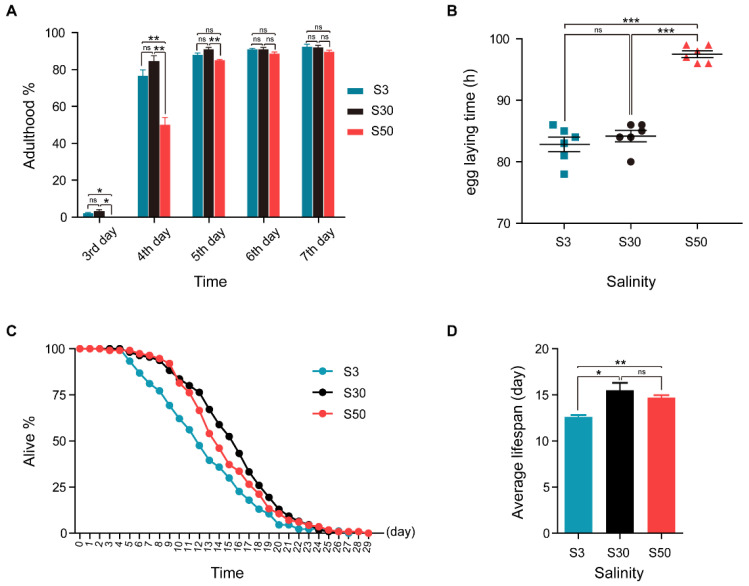
Developmental characterization and lifespan of *L. marina* acclimated to different salinity conditions. (**A**) For developmental analysis, 100 newly hatched L1s were transferred onto each indicated 35 mm dimeter agar plate seeded with 30 μL OP50. The number of adult worms was scored every day. (**B**) Egg laying time of *L. marina* acclimated to different salinity conditions. One hundred newly hatched L1s were transferred onto each indicated 35 mm dimeter agar plate seeded with 30 μL OP50. The earliest observed egg laying time on each plate was recorded. Six replicates were performed for each experimental condition. (**C**) For lifespan assay, 40 L4 females were transferred to each assay plate, incubated at 20 °C. The number of live and dead worms was determined using a dissecting microscope every 24 h. Live worms were transferred to fresh OP50-seeded plates daily. Three replicates were analyzed. Worms were scored as dead if no response was detected after prodding with a platinum wire. Dead worms on the wall of the plate were not counted. (**D**) Average lifespan of *L. marina* acclimated to different salinity conditions. Error bars represent the standard error of the mean from replicated experiments. Differences between groups were analyzed statistically employing the two-tailed Student’s *t*-test. *p* < 0.05 was considered statistically significant. * *p* < 0.05, ** *p* < 0.01, *** *p* < 0.001, ns—not significant.

**Figure 2 genes-13-00651-f002:**
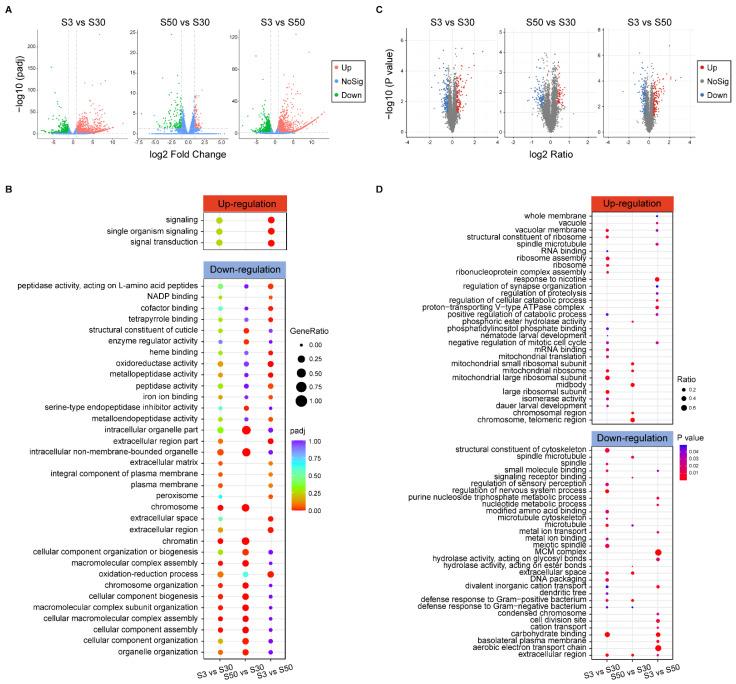
Differentially expressed genes (DEGs) and proteins (DEPs) identified via transcriptomic and proteomic analysis of L1 marine nematodes acclimated to low and high salinity conditions. (**A**) Volcano plot of identified DEGs in different comparison groups via RNA-seq analysis (|log_2_foldchange| > 1; DESeq2 padj < 0.05 was set as the differential gene screening threshold). Red dots (Up) represent significantly up-regulated genes, green dots (Down) represent significantly down-regulated genes, blue dots (NoSig) represent insignificantly differentially expressed genes. (**B**) GO enrichment analysis for DEGs. |log_2_foldchange| > 1; DESeq2 padj < 0.05 was set as the differential gene screening threshold. GO enrichment analysis of DEGs was achieved using the clusterProfiler R package (v3.4.4), an adjusted *p* value (padj) < 0.05 was considered significantly enriched. The color from red to purple represents the significance of the enrichment. GeneRatio was defined as the ratio of the number of differential genes annotated to the GO term to the total number of differential genes. (**C**) Volcano plot of identified DEPs in different comparison groups via proteomic analysis (Ratio of fold change > 1.3 or < 1/1.3; *p* value < 0.05 was set as the differential protein screening threshold). Red dots (Up) represent significantly up-regulated proteins, blue dots (Down) represent significantly down-regulated proteins, grey dots (NoSig) represent insignificantly differentially expressed proteins. (**D**) GO enrichment analysis for DEPs. Ratio of fold change > 1.3 or < 1/1.3; *p* value < 0.05 was set as the differential protein screening threshold. A corrected *p* value < 0.05 was considered significantly enriched. The color represents the significance of the enrichment. Ratio was defined as the ratio of the number of differential proteins annotated to the GO term to the number of background proteins.

**Figure 3 genes-13-00651-f003:**
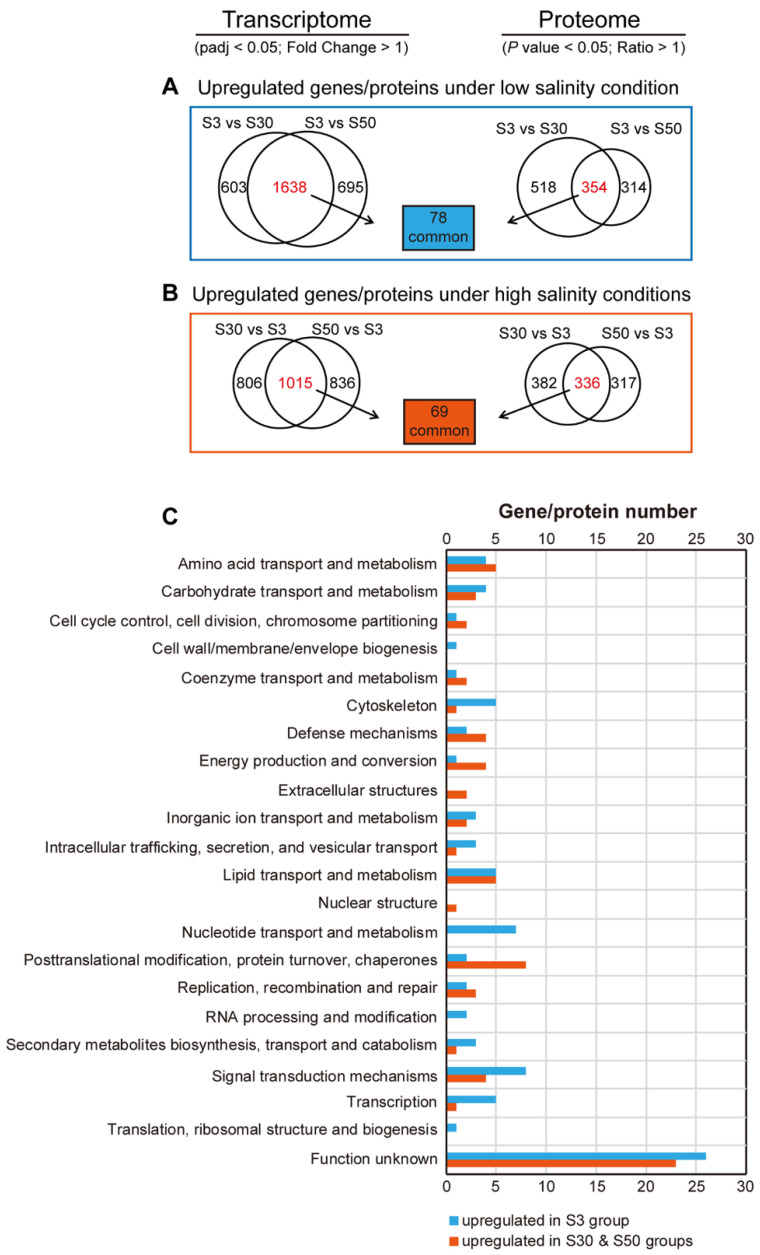
Identification of genes expressed consistently at both mRNA and protein levels under low- and high-salinity conditions. (**A**) Identification of 78 genes with up-regulated expression at both mRNA and protein levels under the low-salinity condition (S3 group) compared with high-salinity conditions (S30 and S50 groups). DESeq2 padj < 0.05; fold change > 1 was set as the differential gene screening threshold. *p* value < 0.05; Ratio > 1 was set as the differential protein screening threshold. (**B**) Identification of 69 genes with up-regulated expression at both mRNA and protein levels under high-salinity conditions (S30 and S50 groups) compared with the low-salinity condition (S3 group). DESeq2 padj < 0.05; fold change > 1 was set as the differential gene screening threshold. *p* value < 0.05; Ratio > 1 was set as the differential protein screening threshold. (**C**) Classification of differential genes expressed under low- and high-salinity conditions based on COG annotation.

**Figure 4 genes-13-00651-f004:**
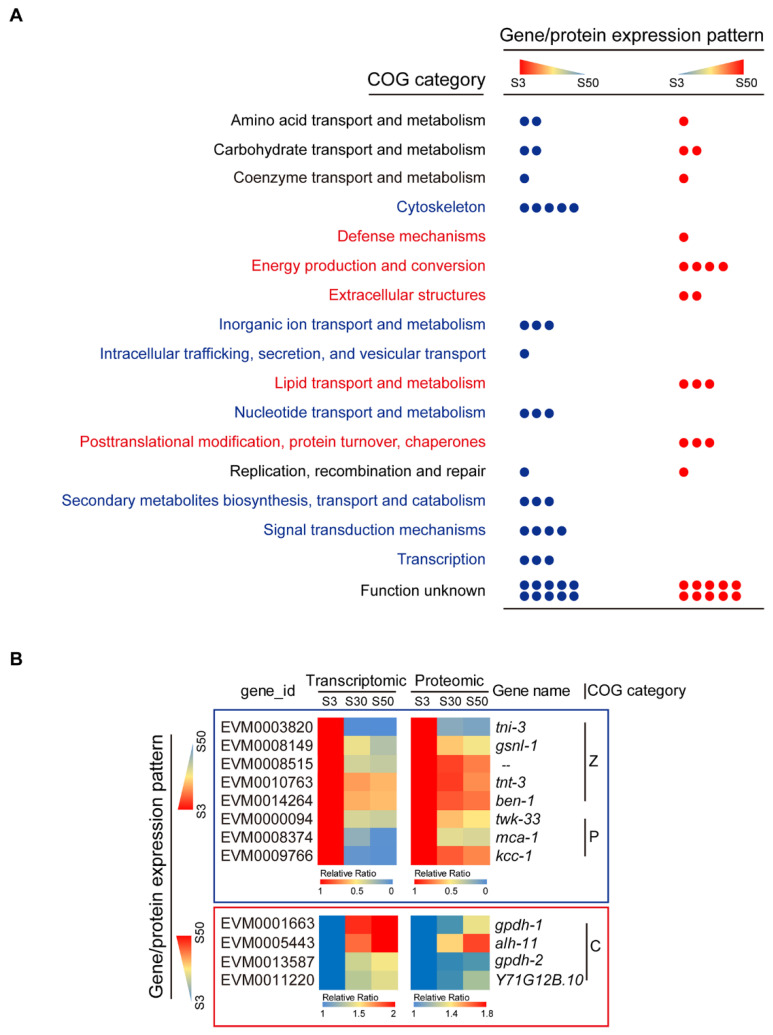
Identification of genes and their corresponding proteins whose abundance was directly or inversely proportional to environmental salinity. (**A**) Differential genes that were expressed in a salinity-dependent pattern were classified based on COG annotation. Each dot represents an individual gene. Blue: genes/proteins that were down-regulated in expression with increasing salinity. Red: genes/proteins that were up-regulated in expression with increasing salinity. Detailed information can be found in Appendix A. (**B**) Expression profiles of representative genes/proteins with the opposite pattern.

**Table 1 genes-13-00651-t001:** Detailed information for 78 common DEGs and DEPs up-regulated specifically under low salinity.

Gene_id	Transcriptomic Level				Proteomic Level				Gene Name	
	S3 vs. S30		S3 vs. S50		S3 vs. S30		S3 vs. S50			
	Fold Change	padj	Fold Change	padj	Ratio	*p* Value	Ratio	*p* Value		
Amino acid transport and metabolism [E]										
EVM0001780	2.25	1.84 × 10^−13^	2.52	8.61 × 10^−17^	1.26	1.64 × 10^−2^	1.28	2.08 × 10^−2^	*gdh-1*	*
EVM0014029	1.59	2.17 × 10^−2^	2.14	6.98 × 10^−5^	1.15	1.32 × 10^−2^	1.10	3.28 × 10^−2^	*Y51F10.4*	
EVM0015459	3.72	9.06 × 10^−23^	3.13	7.30 × 10^−13^	1.32	6.64 × 10^−4^	1.43	8.14 × 10^−3^	*tatn-1*	*
EVM0016867	27.80	6.84 × 10^−9^	257.43	4.23 × 10^−9^	1.25	3.13 × 10^−4^	1.28	2.62 × 10^−3^	*ltah-1.1*	
Carbohydrate transport and metabolism [G]										
EVM0004057	8.37	5.63 × 10^−118^	6.22	3.31 × 10^−74^	1.67	1.87 × 10^−4^	1.81	5.49 × 10^−5^	*F08F3.4*	*
EVM0009122	1.96	6.03 × 10^−3^	2.74	1.56 × 10^−6^	1.63	3.60 × 10^−4^	2.00	7.89 × 10^−5^	*aman-1*	
EVM0012344	1.89	6.04 × 10^−4^	2.05	2.85 × 10^−4^	3.24	2.79 × 10^−4^	5.88	8.68 × 10^−5^	*bus-2*	*
Coenzyme transport and metabolism [H]										
EVM0003915	7.82	5.50 × 10^−36^	10.40	3.53 × 10^−58^	2.06	1.26 × 10^−3^	2.59	9.80 × 10^−4^	*amx-3*	
Cytoskeleton [Z]										
EVM0003820	139.48	1.27 × 10^−116^	8007.93	3.44 × 10^−25^	6.48	5.34 × 10^−6^	8.40	5.33 × 10^−5^	*tni-3*	
EVM0008149	2.25	9.45 × 10^−7^	3.58	6.89 × 10^−16^	1.73	4.48 × 10^−5^	2.12	4.03 × 10^−4^	*gsnl-1*	
EVM0008515	2.71	4.75 × 10^−4^	3.05	4.50 × 10^−5^	1.16	2.10 × 10^−2^	1.36	2.66 × 10^−3^	*--*	
EVM0010763	1.50	2.98 × 10^−2^	1.63	1.72 × 10^−2^	1.14	7.28 × 10^−3^	1.42	2.68 × 10^−4^	*tnt-3*	
EVM0014264	1.59	1.54 × 10^−3^	1.67	7.27 × 10^−4^	1.23	1.80 × 10^−3^	1.33	2.29 × 10^−4^	*ben-1*	
Defense mechanisms [V]										
EVM0008469	1.32	6.62 × 10^−3^	1.50	8.32 × 10^−4^	1.23	9.30 × 10^−3^	1.16	4.24 × 10^−2^	*vhp-1*	
Energy production and conversion [C]										
EVM0007934	36.01	6.99 × 10^−70^	35.72	3.93 × 10^−62^	2.07	9.81 × 10^−4^	1.95	1.12 × 10^−3^	*vha-5*	*
Inorganic ion transport and metabolism [P]										
EVM0000094	2.61	1.20 × 10^−3^	2.87	1.89 × 10^−5^	1.67	3.37 × 10^−4^	1.95	1.25 × 10^−4^	*twk-33*	*
EVM0008374	5.51	1.11 × 10^−9^	34.21	1.11 × 10^−44^	2.38	4.13 × 10^−4^	2.58	3.86 × 10^−4^	*mca-1*	*
EVM0009766	19.15	2.94 × 10^−2^	38.15	6.00 × 10^−3^	1.24	4.01 × 10^−2^	1.39	1.33 × 10^−2^	*kcc-1*	
Intracellular trafficking, secretion, and vesicular transport [U]										
EVM0000891	2.10	2.44 × 10^−9^	1.97	8.13 × 10^−7^	1.16	7.18 × 10^−3^	1.21	5.46 × 10^−3^	*rab-7*	
EVM0013621	1.33	7.43 × 10^−4^	1.49	9.89 × 10^−4^	1.09	4.34 × 10^−2^	1.15	4.15 × 10^−3^	*atg-2*	
EVM0016452	2.56	7.30 × 10^−10^	2.47	1.36 × 10^−9^	1.11	2.44 × 10^−2^	1.17	8.78 × 10^−3^	*tps-2*	*
Lipid transport and metabolism [I]										
EVM0006214	2.90	5.00 × 10^−6^	1.87	1.54 × 10^−2^	1.16	1.98 × 10^−2^	1.21	1.13 × 10^−2^	*Y71F9B.9*	*
EVM0014613	1.91	6.07 × 10^−7^	1.69	5.75 × 10^−4^	1.12	2.86× 10^−2^	1.21	8.41 × 10^−3^	*cest-26*	
EVM0016161	2.08	9.18 × 10^−8^	1.63	1.71 × 10^−3^	1.41	2.60 × 10^−3^	1.35	3.67 × 10^−3^	*C01B10.3*	*
EVM0016904	2.36	1.94 × 10^−12^	2.16	3.75 × 10^−9^	1.20	3.50 × 10^−2^	1.16	1.75 × 10^−2^	*faah-2*	
Nucleotide transport and metabolism [F]										
EVM0002534	2.15	1.97 × 10^−9^	2.37	3.56 × 10^−9^	1.09	2.85 × 10^−3^	1.05	2.18 × 10^−2^	*pyr-1*	
EVM0007122	2.67	5.94 × 10^−7^	2.85	2.32 × 10^−7^	1.23	8.96 × 10^−3^	1.35	1.36 × 10^−2^	*alh-3*	
EVM0007619	2.08	1.00 × 10^−4^	1.99	4.62 × 10^−4^	1.14	8.02 × 10^−4^	1.11	2.43 × 10^−4^	*Y43F8C.13*	*
EVM0010113	3.05	8.69 × 10^−29^	3.66	7.64 × 10^−23^	1.22	1.15 × 10^−2^	1.24	1.39 × 10^−2^	*Y48A6B.7*	
EVM0012708	1.45	2.49 × 10^−2^	1.72	2.07 × 10^−3^	1.20	1.64 × 10^−4^	1.18	4.23 × 10^−6^	*hint-3*	
EVM0015677	1.62	1.89 × 10^−4^	2.05	7.44 × 10^−7^	1.10	5.34 × 10^−4^	1.30	1.06 × 10^−5^	*ent-1*	
EVM0017439	9.94	1.78 × 10^−38^	3.97	1.50 × 10^−14^	1.28	1.09 × 10^−5^	1.25	2.19 × 10^−5^	*pnp-1*	*
Posttranslational modification, protein turnover, chaperones [O]										
EVM0013078	1.50	4.08 × 10^−4^	1.39	3.00 × 10^−2^	1.19	1.17 × 10^−3^	1.17	7.83 × 10^−3^	*ahsa-1*	
Replication, recombination and repair [L]										
EVM0003459	1.44	3.58 × 10^−2^	1.60	1.03 × 10^−2^	1.30	2.87 × 10^−4^	1.29	2.26 × 10^−3^	*F21D5.5*	
EVM0012370	1.56	4.40 × 10^−2^	1.65	1.74 × 10^−2^	1.25	4.57 × 10^−2^	1.31	2.14 × 10^−2^	*Y87G2A.19*	
RNA processing and modification [A]										
EVM0009834	2.16	2.64 × 10^−8^	2.20	1.72 × 10^−8^	1.50	2.64 × 10^−3^	1.36	2.48 × 10^−2^	*dxbp-1*	
EVM0002279	1.70	5.20 × 10^−3^	1.69	3.08 × 10^−3^	1.10	3.91 × 10^−2^	1.11	2.51 × 10^−2^	*usip-1*	
Secondary metabolites biosynthesis, transport and catabolism [Q]										
EVM0004846	3.40	3.73 × 10^−6^	3.62	4.03 × 10^−7^	1.21	1.91 × 10^−2^	1.25	2.88 × 10^−4^	*amx-2*	
EVM0008743	2.93	4.91 × 10^−2^	3.41	2.05 × 10^−2^	1.17	1.66 × 10^−3^	1.26	1.39 × 10^−2^	*R04B5.5*	*
EVM0008816	143.66	3.37 × 10^−39^	2394.32	1.13 × 10^−18^	1.75	7.32 × 10^−5^	1.86	6.57 × 10^−4^	*R05D8.7*	
Signal transduction mechanisms [T]										
EVM0002057	2.13	1.86 × 10^−5^	3.40	4.49 × 10^−11^	1.25	3.08 × 10^−3^	1.38	1.47 × 10^−3^	*Y57A10A.26*	*
EVM0002262	8.81	7.62 × 10^−21^	6.65	7.91 × 10^−20^	1.13	1.46 × 10^−3^	1.10	5.17 × 10^−3^	*F38A5.2*	
EVM0003443	1.44	4.26 × 10^−4^	1.73	1.48 × 10^−5^	1.19	1.08 × 10^−3^	1.30	8.60 × 10^−5^	*mnk-1*	
EVM0003881	1.36	7.08 × 10^−3^	1.62	5.01 × 10^−5^	1.12	2.07 × 10^−3^	1.14	8.04 × 10^−3^	*ppfr-1*	
EVM0006886	1.89	3.05 × 10^−8^	2.61	3.38 × 10^−21^	1.19	4.70 × 10^−2^	1.17	9.03 × 10^−3^	*T19D2.2*	
EVM0009042	1.65	3.96 × 10^−3^	2.10	9.24 × 10^−5^	1.14	2.44 × 10^−2^	1.18	1.36 × 10^−2^	*tnc-2*	
EVM0012157	1.45	3.60 × 10^−3^	1.35	3.70 × 10^−2^	1.09	4.47 × 10^−3^	1.11	3.38 × 10^−3^	*arr-1*	
Transcription [K]										
EVM0002661	1.34	6.86 × 10^−3^	1.39	1.18 × 10^−2^	1.21	6.71 × 10^−4^	1.14	1.24 × 10^−2^	*ldb-1*	
EVM0004245	48.12	2.54 × 10^−33^	137.83	7.64 × 10^−23^	1.09	8.29 × 10^−3^	1.04	5.04 × 10^−3^	*sta-1*	
EVM0011147	1.47	1.82 × 10^−4^	1.74	5.70 × 10^−6^	1.26	7.41 × 10^−3^	1.27	2.21 × 10^−2^	*isw-1*	
EVM0011991	1.36	2.63 × 10^−2^	1.39	2.06 × 10^−2^	1.35	2.19 × 10^−2^	1.39	6.50 × 10^−3^	*gmeb-2*	
EVM0015999	4.12	4.12 × 10^−26^	15.20	5.43 × 10^−63^	1.29	1.46 × 10^−2^	1.57	8.14 × 10^−4^	*dmd-7*	*
Function unknown [S]										
EVM0001362	1.54	2.88 × 10^−3^	5.71	1.21 × 10^−13^	2.61	1.66 × 10^−4^	4.27	6.79 × 10^−5^	*--*	*
EVM0001899	2.14	1.43 × 10^−6^	1.79	1.89 × 10^−5^	1.35	2.53 × 10^−3^	1.33	4.27 × 10^−2^	*anmt-3*	
EVM0002515	2.44	1.66 × 10^−7^	2.26	4.43 × 10^−6^	1.49	8.70 × 10^−4^	1.71	4.57 × 10^−3^	*R02C2.7*	*
EVM0002519	170.92	1.25 × 10^−44^	131.09	1.34 × 10^−50^	1.31	3.64 × 10^−2^	2.24	7.07 × 10^−4^	*lfi-1*	
EVM0002991	10.48	1.66 × 10^−30^	137.72	4.88 × 10^−31^	2.81	6.07 × 10^−5^	2.73	6.24 × 10^−4^	*rop-1*	
EVM0003311	1.72	1.98 × 10^−5^	1.38	1.26 × 10^−2^	1.29	9.94 × 10^−4^	1.49	6.50 × 10^−3^	*C23H3.2*	*
EVM0003972	16.96	1.75 × 10^−15^	14.33	2.22 × 10^−15^	1.72	2.28 × 10^−3^	2.60	2.65 × 10^−4^	*ttr-15*	*
EVM0004638	6.15	1.64 × 10^−8^	3.28	5.32 × 10^−5^	1.52	1.13 × 10^−2^	1.68	1.49 × 10^−3^	*ttr-30*	*
EVM0004934	1.53	7.09 × 10^−4^	1.56	4.59 × 10^−3^	1.08	5.46 × 10^−3^	1.13	1.56 × 10^−3^	*arrd-25*	
EVM0005485	2.18	2.02 × 10^−5^	2.50	1.33 × 10^−5^	1.66	2.46 × 10^−2^	1.61	1.60 × 10^−2^	*marg-1*	
EVM0006229	1.41	1.09 × 10^−2^	1.39	4.23 × 10^−2^	1.12	1.79 × 10^−2^	1.21	2.57 × 10^−3^	*Y9C12A.1*	
EVM0006956	3.36	2.33 × 10^−5^	3.34	5.43 × 10^−6^	1.46	5.03 × 10^−4^	1.52	3.68 × 10^−4^	*F36G3.2*	*
EVM0009105	1.70	1.34 × 10^−3^	1.85	7.32 × 10^−4^	1.10	1.78 × 10^−2^	1.15	2.58 × 10^−3^	*C44E4.5*	*
EVM0009106	6.06	2.69 × 10^−31^	4.66	1.98 × 10^−17^	1.27	5.78 × 10^−3^	1.42	1.30 × 10^−4^	*F46C5.10*	
EVM0009178	1.46	1.25 × 10^−2^	1.60	2.53 × 10^−3^	1.35	2.53 × 10^−4^	1.37	1.07 × 10^−3^	*unc-132*	
EVM0009923	136.66	1.06 × 10^−6^	139.74	8.88 × 10^−7^	1.26	6.69 × 10^−4^	1.18	1.29 × 10^−2^	*T09A5.15*	
EVM0010567	5.09	5.66 × 10^−20^	4.13	1.91 × 10^−12^	1.67	1.22 × 10^−2^	2.58	6.63 × 10^−5^	*F52E1.14*	*
EVM0011338	1.79	2.11 × 10^−4^	2.04	3.41 × 10^−4^	1.99	1.20 × 10^−3^	3.23	2.79 × 10^−5^	*--*	*
EVM0011489	1.98	3.28 × 10^−12^	2.41	1.50 × 10^−15^	1.79	3.31 × 10^−6^	1.84	1.54 × 10^−2^	*K07C11.8*	*
EVM0011529	1.31	9.28 × 10^−3^	1.97	9.31 × 10^−8^	1.52	3.60 × 10^−3^	1.90	6.25 × 10^−4^	*sfxn-1.2*	*
EVM0012853	71.98	1.88 × 10^−13^	49.06	2.03 × 10^−15^	1.26	2.84 × 10^−3^	1.36	3.53 × 10^−2^	*R02F2.7*	
EVM0013037	2.31	3.43 × 10^−5^	1.54	2.04 × 10^−2^	1.21	1.98 × 10^−2^	1.33	1.44 × 10^−2^	*T16G1.6*	*
EVM0014110	87.65	7.07 × 10^−6^	174.53	1.38 × 10^−7^	1.21	2.72 × 10^−2^	1.17	2.84 × 10^−2^	*F53F1.2*	
EVM0014277	1.92	3.57 × 10^−2^	2.64	7.83 × 10^−4^	1.38	2.81 × 10^−3^	1.54	1.56 × 10^−2^	*F36G3.2*	*
EVM0017018	2.85	1.50 × 10^−20^	4.67	5.31 × 10^−31^	1.28	1.12 × 10^−2^	1.61	8.47 × 10^−4^	*far-8*	
EVM0017308	6746.66	1.14 × 10^−24^	6896.64	1.92 × 10^−24^	3.83	4.29 × 10^−5^	4.29	4.94 × 10^−5^	*T02B11.3*	*

Note: Genes marked with a red asterisk can be specifically induced by low-salinity stress, as shown in Appendix A analyzed based on data from our previous study [39].

**Table 2 genes-13-00651-t002:** Detailed information of 69 common DEGs and DEPs up-regulated specifically under high-salinity environments.

Gene_id	Transcriptomic Level				Proteomic Level				Gene Name	
	S30 vs. S3		S50 vs. S3		S30 vs. S3		S50 vs. S3			
	Fold Change	padj	Fold Change	padj	Ratio	*p* Value	Ratio	*p* Value		
Amino acid transport and metabolism [E]										
EVM0002320	3.53	5.01 × 10^−8^	2.66	1.69 × 10^−4^	1.48	1.04 × 10^−2^	1.29	2.62 × 10^−3^	*T03D8.6*	
EVM0004826	1.74	4.23 × 10^−4^	3.19	4.57 × 10^−15^	1.48	3.17 × 10^−3^	1.37	1.10 × 10^−2^	*Y32F6A.4*	
EVM0006400	2.23	2.04 × 10^−18^	2.33	1.08 × 10^−10^	1.72	6.00 × 10^−5^	1.93	6.37 × 10^−7^	*alh-6*	
EVM0007127	1.54	1.32 × 10^−6^	2.06	2.21 × 10^−11^	1.24	1.93 × 10^−2^	1.17	1.84 × 10^−2^	*alh-9*	
Carbohydrate transport and metabolism [G]										
EVM0013041	1.34	1.49 × 10^−2^	1.59	5.87 × 10^−4^	1.24	1.61 × 10^−2^	1.63	6.94 × 10^−5^	*ttx-7*	
EVM0013408	2.12	2.27 × 10^−9^	3.74	1.73 × 10^−14^	1.25	1.38 × 10^−2^	1.26	6.31 × 10^−4^	*F25A2.1*	
EVM0017389	1.90	1.49 × 10^−2^	2.70	2.24 × 10^−6^	1.31	2.35 × 10^−2^	1.30	1.79 × 10^−2^	*aagr-1*	
Cell cycle control, cell division, chromosome partitioning [D]										
EVM0001567	1.78	1.55 × 10^−3^	1.85	5.91 × 10^−4^	1.15	2.69 × 10^−3^	1.06	2.79 × 10^−2^	*mua-6*	
EVM0007071	2.04	2.28 × 10^−5^	1.89	6.65 × 10^−4^	1.05	5.38 × 10^−4^	1.16	1.39 × 10^−3^	*W01C8.5*	
Coenzyme transport and metabolism [H]										
EVM0013159	3.82	1.80 × 10^−13^	4.08	1.39 × 10^−15^	1.29	7.43 × 10^−4^	1.57	1.11 × 10^−3^	*F37C4.6*	*
EVM0015524	1.39	3.80 × 10^−2^	1.43	3.60 × 10^−2^	1.14	1.22 × 10^−2^	1.11	7.32 × 10^−3^	*--*	
Cytoskeleton [Z]										
EVM0000074	2.29	1.84 × 10^−7^	2.19	2.89 × 10^−6^	1.42	6.68 × 10^−5^	1.22	2.34 × 10^−3^	*tba-4*	
Defense mechanisms [V]										
EVM0001236	7.77	6.41 × 10^−25^	6.66	2.80 × 10^−14^	1.96	1.62 × 10^−3^	1.40	3.10 × 10^−2^	*clec-49*	
EVM0008192	3.56	9.74 × 10^−15^	3.92	6.69 × 10^−15^	1.66	1.67 × 10^−4^	1.38	9.47 × 10^−3^	*clec-48*	*
EVM0009848	1.55	1.22 × 10^−4^	2.07	1.26 × 10^−6^	1.10	4.69 × 10^−2^	1.12	5.00 × 10^−2^	*cri-3*	
EVM0017657	1.41	3.58 × 10^−2^	1.82	1.35 × 10^−4^	1.18	8.56 × 10^−3^	1.12	1.05 × 10^−2^	*mpst-1*	
Energy production and conversion [C]										
EVM0001663	1.91	5.76 × 10^−5^	2.70	3.12 × 10^−10^	1.11	1.72 × 10^−2^	1.37	9.53 × 10^−4^	*gpdh-1*	*
EVM0005443	1.77	2.02 × 10^−5^	2.37	2.88 × 10^−8^	1.44	1.94 × 10^−4^	1.68	2.47 × 10^−4^	*alh-11*	
EVM0011220	1.36	1.09 × 10^−2^	1.44	1.49 × 10^−2^	1.10	3.19 × 10^−2^	1.26	3.36 × 10^−3^	*Y71G12B.10*	*
EVM0013587	1.40	1.59 × 10^−3^	1.48	1.39 × 10^−3^	1.08	1.10 × 10^−2^	1.12	3.67 × 10^−3^	*gpdh-2*	
Extracellular structures [W]										
EVM0011248	2.13	1.60× 10^−12^	2.44	2.02 × 10^−16^	1.60	6.06 × 10^−4^	1.79	2.89 × 10^−4^	*ost-1*	
EVM0013254	1.42	4.10 × 10^−2^	2.09	5.10 × 10^−6^	1.20	5.24 × 10^−3^	1.44	1.10 × 10^−3^	*col-160*	*
Inorganic ion transport and metabolism [P]										
EVM0000243	1.95	2.34 × 10^−3^	1.82	4.86 × 10^−3^	1.38	1.59 × 10^−3^	1.28	1.31 × 10^−2^	*snf-9*	*
EVM0009288	1.26	2.60 × 10^−2^	1.68	6.71 × 10^−5^	1.72	4.17 × 10^−4^	1.57	8.33 × 10^−4^	*aat-4*	
Intracellular trafficking, secretion, and vesicular transport [U]										
EVM0008733	7.63	3.68 × 10^−61^	8.14	3.90 × 10^−32^	1.61	8.18 × 10^−3^	1.38	2.14 × 10^−2^	*nex-2*	
EVM0015971	2.32	9.98 × 10^−5^	1.96	3.76 × 10^−3^	1.34	1.36 × 10^−2^	1.30	2.32 × 10^−2^	*Y50D4A.1*	
Lipid transport and metabolism [I]										
EVM0004076	2.13	1.09 × 10^−5^	4.37	2.62 × 10^−18^	1.54	9.47 × 10^−5^	1.80	3.00 × 10^−5^	*acs-7*	
EVM0009809	2.11	2.78 × 10^−3^	2.59	1.23 × 10^−5^	1.12	2.83 × 10^−2^	1.28	1.82 × 10^−3^	*acox-1.6*	
EVM0011984	1.70	2.10 × 10^−2^	3.04	4.27 × 10^−8^	1.23	2.16 × 10^−4^	1.22	1.75 × 10^−3^	*T20B3.1*	
EVM0012282	2.23	9.01 × 10^−11^	2.26	4.81 × 10^−8^	1.22	6.27 × 10^−3^	1.20	1.22 × 10^−2^	*Y25C1A.13*	
EVM0016519	1.96	1.44 × 10^−5^	2.56	7.18 × 10^−17^	1.57	1.02 × 10^−2^	1.71	4.47 × 10^−3^	*ges-1*	
Posttranslational modification, protein turnover, chaperones [O]										
EVM0002002	2.64	5.00 × 10^−6^	3.29	2.15 × 10^−8^	1.44	4.95 × 10^−3^	1.46	2.50 × 10^−3^	*rle-1*	
EVM0003227	1.59	3.99 × 10^−3^	1.67	2.75 × 10^−3^	1.26	9.02 × 10^−4^	1.33	1.54 × 10^−5^	*F44E7.4*	
EVM0003273	3.27	9.97 × 10^−16^	2.45	1.30 × 10^−7^	1.28	3.87 × 10^−4^	1.21	1.16 × 10^−2^	*cpr-5*	
EVM0005041	2.61	4.92 × 10^−12^	3.08	3.01 × 10^−12^	1.44	1.96 × 10^−2^	1.22	2.19 × 10^−2^	*C35B1.5*	
EVM0005930	2.66	1.03 × 10^−12^	3.02	1.89 × 10^−16^	1.34	2.74 × 10^−2^	1.47	4.03 × 10^−3^	*gst-8*	
EVM0008252	2.56	1.71 × 10^−16^	2.68	4.85 × 10^−13^	1.74	1.89 × 10^−2^	1.50	1.49 × 10^−2^	*fkb-2*	
EVM0008830	1.58	2.22 × 10^−5^	1.61	2.18 × 10^−4^	1.08	3.29 × 10^−2^	1.13	8.10 × 10^−3^	*Y71H2AM.1*	
EVM0014265	1.30	2.24 × 10^−2^	1.78	9.72 × 10^−7^	1.32	2.91 × 10^−2^	1.24	4.66 × 10^−2^	*dpy-31*	
Replication, recombination and repair [L]										
EVM0004517	2.04	3.41 × 10^−4^	1.66	3.34 × 10^−2^	1.20	2.67 × 10^−2^	1.31	1.13 × 10^−3^	*mcm-7*	
EVM0013043	2.09	6.08 × 10^−3^	2.23	1.26 × 10^−3^	1.19	4.98 × 10^−2^	1.30	6.29 × 10^−5^	*mcm-4*	
EVM0017031	2.13	2.40 × 10^−4^	1.81	9.07 × 10^−3^	1.15	2.83 × 10^−2^	1.31	8.34 × 10^−4^	*mcm-3*	
Secondary metabolites biosynthesis, transport and catabolism [Q]										
EVM0008044	3.25	2.55 × 10^−21^	3.26	6.19 × 10^−12^	1.35	1.61 × 10^−2^	1.24	6.73 × 10^−3^	*comt-3*	
Signal transduction mechanisms [T]										
EVM0007854	10.24	1.74 × 10^−2^	9.18	1.42 × 10^−2^	2.39	5.95 × 10^−4^	2.66	1.82 × 10^−4^	*H10E21.4*	
EVM0008918	1.60	1.56 × 10^−5^	1.49	5.95 × 10^−4^	1.19	6.35 × 10^−4^	1.14	8.33 × 10^−6^	*lrp-1*	*
Transcription [K]										
EVM0014859	1.66	7.09 × 10^−3^	1.62	4.20 × 10^−3^	1.19	8.92 × 10^−4^	1.11	2.70 × 10^−2^	*sta-2*	
Function unknown [S]										
EVM0000249	1.66	2.42 × 10^−2^	2.10	2.34 × 10^−4^	1.40	4.12 × 10^−3^	1.15	1.16 × 10^−2^	*--*	
EVM0000791	1.92	1.77 × 10^−3^	2.86	1.06 × 10^−4^	1.32	3.06 × 10^−2^	1.59	7.65 × 10^−4^	*bigr-1*	
EVM0001976	1.79	2.94 × 10^−12^	1.52	3.09 × 10^−3^	1.62	2.53 × 10^−2^	1.28	8.45 × 10^−3^	*far-1*	
EVM0004494	1.36	3.68 × 10^−2^	1.87	2.75 × 10^−7^	1.24	6.34 × 10^−5^	1.24	1.35 × 10^−3^	*F47G3.1*	
EVM0004595	2.28	5.74 × 10^−3^	4.69	1.53 × 10^−12^	1.17	7.26 × 10^−3^	1.46	8.77 × 10^−4^	*F53F1.2*	
EVM0005666	1.78	4.90 × 10^−3^	2.67	1.16 × 10^−6^	1.09	1.45 × 10^−2^	1.23	1.80 × 10^−4^	*bath-3*	
EVM0005758	1.39	3.91 × 10^−2^	1.48	2.03 × 10^−2^	1.11	1.41 × 10^−3^	1.15	3.84 × 10^−2^	*--*	
EVM0006660	1.24	3.26 × 10^−2^	1.97	1.05 × 10^−9^	1.19	6.42 × 10^−3^	1.31	3.06 × 10^−4^	*abcf-2*	
EVM0006900	1.80	4.62 × 10^−7^	2.22	2.13 × 10^−10^	2.19	6.66 × 10^−5^	2.91	2.36 × 10^−5^	*F45E1.4*	
EVM0008841	4.75	6.94 × 10^−10^	3.32	5.93 × 10^−10^	1.74	4.43 × 10^−6^	1.35	1.70 × 10^−4^	*--*	
EVM0009309	1.65	3.81 × 10^−3^	2.22	4.98 × 10^−6^	1.24	1.08 × 10^−4^	1.20	3.27 × 10^−3^	*clec-62*	*
EVM0009573	1.90	1.55 × 10^−3^	1.80	7.17 × 10^−3^	1.19	2.89 × 10^−2^	1.51	1.09 × 10^−3^	*plin-1*	
EVM0010240	1.39	3.55 × 10^−3^	1.39	1.65 × 10^−2^	1.12	7.53 × 10^−3^	1.20	2.92 × 10^−3^	*wdr-20*	
EVM0011170	1.86	4.54 × 10^−2^	1.80	4.45 × 10^−2^	1.89	1.03 × 10^−2^	1.68	1.31 × 10^−3^	*ttr-27*	
EVM0011921	2.13	6.49 × 10^−9^	2.23	5.65 × 10^−10^	1.76	5.86 × 10^−3^	1.29	2.93 × 10^−2^	*F47G3.4*	*
EVM0012075	1.67	1.60 × 10^−2^	2.93	1.91 × 10^−8^	1.45	1.22 × 10^−3^	1.63	8.95 × 10^−3^	*T10B10.4*	*
EVM0012459	2.35	4.58 × 10^−5^	2.38	1.22 × 10^−4^	1.25	2.19 × 10^−3^	1.08	4.78 × 10^−3^	*Y12A6A.1*	*
EVM0013261	2.11	6.67 × 10^−11^	2.13	1.72 × 10^−6^	1.43	3.62 × 10^−3^	1.12	3.50 × 10^−2^	*F42A10.7*	
EVM0013349	1.61	9.16 × 10^−4^	1.73	2.27 × 10^−4^	1.15	1.31 × 10^−2^	1.33	1.40 × 10^−2^	*--*	
EVM0014431	1.46	2.47 × 10^−3^	1.42	9.95 × 10^−3^	1.05	1.57 × 10^−2^	1.13	2.88 × 10^−2^	*such-1*	
EVM0014451	14.39	3.33 × 10^−75^	7.00	2.51 × 10^−15^	1.81	1.47 × 10^−2^	1.56	4.02 × 10^−3^	*F57H12.6*	
EVM0014569	10.18	7.74 × 10^−64^	5.71	4.22 × 10^−21^	1.65	1.34 × 10^−2^	1.55	3.87 × 10^−3^	*C39E9.8*	
EVM0016108	3.45	2.80 × 10^−28^	3.02	5.13 × 10^−14^	1.53	4.21 × 10^−2^	1.44	4.84 × 10^−2^	*F55H12.4*	

Note: Genes marked with a red asterisk can be specifically induced by high-salinity stress, as shown in Appendix A analyzed based on data from our previous study [39].

## Data Availability

The RNA-seq raw data were submitted to the NCBI under the accession code PRJNA778902. All raw proteomics data are available via ProteomeXchange under identifier PXD029671.

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
