# Peer review of "Transcriptomic and Proteomic Analysis of Marine Nematode Litoditis marina Acclimated to Different Salinities"

_genes, 2022, doi:10.3390/genes13040651_

Round 1

Reviewer 1 Report

Very minor edits:

1- line 21: which might

2- Line 43: "It is extensively known" - change to "it is known"

3- Line 51: "has been reported contributing" - change to "has been reported to contribute"

4- Line 68: change first sentence to "We have previously identified  a broad range of salinity responding genes in L. marina when challenged with either low or high salinity stresses"

5- Line 75: please omit "in the present study"

6- Line 135: Following "the" manufacturer's instructions

7- Line 242: change the word "exhibited" to "was observed"

7- Line 527: gpdh-2 was significantly - not were.

Line 337: "1638 genes were demonstrated upregulation" - change it to "1638 genes were upregulated".

Line 519: the sentence is missing a verb

Author Response

Dear Editors,

       We would like to submit a revised manuscript "Transcriptomic and proteomic analysis of marine nematode Litoditis marina acclimated to different salinities" (Manuscript ID: genes-1630146).

       We would like to thank the reviewers for their invaluable comments and suggestions. Below we detail point-by-point response to the reviewers’ comments.

Sincerely yours,
Liusuo Zhang

Liusuo Zhang PhD
Principal Investigator  
Institute of Oceanology
Chinese Academy of Sciences
Biology building(#2 Room 332)
7 Nanhai Road
Qingdao, Shandong, China 266071
Email: lzhang@qdio.ac.cn
Office:  86-532-82898843

Response to Reviewer 1 Comments

Point 1: line 21: which might

Response 1: We would like to thank the reviewer for your invaluable comment, and we have revised accordingly in Line 21.

Point 2: Line 43: "It is extensively known" - change to "it is known"

Response 2: As suggested by the reviewer, we have revised accordingly in Line 43.

Point 3: Line 51: "has been reported contributing" - change to "has been reported to contribute"

Response 3: According to the reviewer’s invaluable comment, we have revised accordingly in Line 51.

Point 4: Line 68: change first sentence to "We have previously identified  a broad range of salinity responding genes in L. marina when challenged with either low or high salinity stresses"

Response 4: We appreciated the reviewer’s invaluable comment, and we have revised accordingly in Line 68.

Point 5: Line 75: please omit "in the present study"

Response 5: As suggested by the reviewer, we have revised accordingly in Line 76 in the revised manuscript.

Point 6: Line 135: Following "the" manufacturer's instructions

Response 6: We appreciated the reviewer’s invaluable comment, and we have revised accordingly in Line 136 in the revised manuscript.

Point 7: Line 242: change the word "exhibited" to "was observed"

Response 7: As suggested by the reviewer, we have revised accordingly in Line 243 in the revised manuscript.

Point 8: Line 527: gpdh-2 was significantly - not were.

Response 8: We appreciated the reviewer’s invaluable comment, and we have revised accordingly in Line 534 in the revised manuscript.

Point 9: Line 337: "1638 genes were demonstrated upregulation" - change it to "1638 genes were upregulated".

Response 9: As suggested by the reviewer, we have revised accordingly in Line 343 in the revised manuscript.

Point 10: Line 519: the sentence is missing a verb

Response 10: We appreciated the reviewer’s invaluable comment, and we have revised accordingly in Line 527 in the revised manuscript.

Response to Reviewer 2 Comments

Point 1: The paper aims to describe for the first time the genome-wide transcriptional and proteomic analysis of the marine nematode L. marina acclimated to either low or high salinity conditions. Overall, the study is very well presented and interesting. However, the work is very long and tiring to read. I advise the authors to delete the conclusions part in the introduction (from line 74) and leave only the purpose of the paper, also reducing the discussion.

Response 1: We appreciated the reviewer’s invaluable comments and suggestions, and we have deleted the conclusions part in the introduction and also reduced certain discussion parts in the revised manuscript.

Point 2: Concerning the M&M and results, the data are well-presented, just I suggest substituting Fig 2, A, B with Volcano plots.

Response 2: As suggested by the reviewer, we revised Fig 2 with Volcano plots accordingly in the current revised manuscript.

Reviewer 2 Report

The paper aims to describe for the first time the genome-wide transcriptional and proteomic analysis of the marine nematode L. marina acclimated to either low or high salinity conditions. Overall, the study is very well presented and interesting. However, the work is very long and tiring to read. I advise the authors to delete the conclusions part in the introduction (from line 74) and leave only the purpose of the paper, also reducing the discussion.

Concerning the M&M and results, the data are well-presented, just I suggest substituting Fig 2, A, B with Volcano plots.

Author Response

Dear Editors,

       We would like to submit a revised manuscript "Transcriptomic and proteomic analysis of marine nematode Litoditis marina acclimated to different salinities"(Manuscript ID: genes-1630146).

       We would like to thank the reviewers for their invaluable comments and suggestions. Below we detail point-by-point response to the reviewers’ comments.

Sincerely yours,
Liusuo Zhang

Liusuo Zhang PhD
Principal Investigator  
Institute of Oceanology
Chinese Academy of Sciences
Biology building(#2 Room 332)
7 Nanhai Road
Qingdao, Shandong, China 266071
Email: lzhang@qdio.ac.cn
Office:  86-532-82898843

Response to Reviewer 1 Comments

Point 1: line 21: which might

Response 1: We would like to thank the reviewer for your invaluable comment, and we have revised accordingly in Line 21.

Point 2: Line 43: "It is extensively known" - change to "it is known"

Response 2: As suggested by the reviewer, we have revised accordingly in Line 43.

Point 3: Line 51: "has been reported contributing" - change to "has been reported to contribute"

Response 3: According to the reviewer’s invaluable comment, we have revised accordingly in Line 51.

Point 4: Line 68: change first sentence to "We have previously identified  a broad range of salinity responding genes in L. marina when challenged with either low or high salinity stresses"

Response 4: We appreciated the reviewer’s invaluable comment, and we have revised accordingly in Line 68.

Point 5: Line 75: please omit "in the present study"

Response 5: As suggested by the reviewer, we have revised accordingly in Line 76 in the revised manuscript.

Point 6: Line 135: Following "the" manufacturer's instructions

Response 6: We appreciated the reviewer’s invaluable comment, and we have revised accordingly in Line 136 in the revised manuscript.

Point 7: Line 242: change the word "exhibited" to "was observed"

Response 7: As suggested by the reviewer, we have revised accordingly in Line 243 in the revised manuscript.

Point 8: Line 527: gpdh-2 was significantly - not were.

Response 8: We appreciated the reviewer’s invaluable comment, and we have revised accordingly in Line 534 in the revised manuscript.

Point 9: Line 337: "1638 genes were demonstrated upregulation" - change it to "1638 genes were upregulated".

Response 9: As suggested by the reviewer, we have revised accordingly in Line 343 in the revised manuscript.

Point 10: Line 519: the sentence is missing a verb

Response 10: We appreciated the reviewer’s invaluable comment, and we have revised accordingly in Line 527 in the revised manuscript.

Response to Reviewer 2 Comments

Point 1: The paper aims to describe for the first time the genome-wide transcriptional and proteomic analysis of the marine nematode L. marina acclimated to either low or high salinity conditions. Overall, the study is very well presented and interesting. However, the work is very long and tiring to read. I advise the authors to delete the conclusions part in the introduction (from line 74) and leave only the purpose of the paper, also reducing the discussion.

Response 1: We appreciated the reviewer’s invaluable comments and suggestions, and we have deleted the conclusions part in the introduction and also reduced certain discussion parts in the revised manuscript.

Point 2: Concerning the M&M and results, the data are well-presented, just I suggest substituting Fig 2, A, B with Volcano plots.

Response 2: As suggested by the reviewer, we revised Fig 2 with Volcano plots accordingly in the current revised manuscript.
